# Implicit Convolutional Kernels for Steerable CNNs

**Maksim Zhdanov**[*]
AMLab, University of Amsterdam
m.zhdanov@uva.nl

**Nico Hoffmann**
Helmholtz-Zentrum
Dresden-Rossendorf

**Gabriele Cesa**
Qualcomm AI Research[†]
AMLab, University of Amsterdam

## Abstract

Steerable convolutional neural networks (CNNs) provide a general framework for building neural networks equivariant to translations and transformations of an origin-preserving group $G$, such as reflections and rotations. They rely on standard convolutions with $G$-steerable kernels obtained by analytically solving the group-specific equivariance constraint imposed onto the kernel space. As the solution is tailored to a particular group $G$, implementing a kernel basis does not generalize to other symmetry transformations, complicating the development of general group equivariant models. We propose using implicit neural representation via multi-layer perceptrons (MLPs) to parameterize $G$-steerable kernels. The resulting framework offers a simple and flexible way to implement Steerable CNNs and generalizes to any group $G$ for which a $G$-equivariant MLP can be built. We prove the effectiveness of our method on multiple tasks, including N-body simulations, point cloud classification and molecular property prediction.

## 1 Introduction

Equivariant deep learning is a powerful tool for high-dimensional problems with known data domain symmetry. By incorporating this knowledge as inductive biases into neural networks, the hypothesis class of functions can be significantly restricted, leading to improved data efficiency and generalization performance [6]. Convolutional neural networks [28] (CNNs) are a prominent example as they are equivariant to translations. Group-equivariant CNNs [9] (G-CNNs) generalize CNNs to exploit a larger number of symmetries via group convolutions, making them equivariant to the desired symmetry group, such as the Euclidean group $E(n)$ that encompasses translations, rotations, and reflections in $n$-dimensional Euclidean space. In physics and chemistry, many important problems, such as molecular modelling or point clouds, rely on the Euclidean group. Objects defined in physical space have properties that are invariant or equivariant to Euclidean transformations, and respecting this underlying symmetry is often desired for the model to perform as expected.

Neural networks can be parameterized in various ways to incorporate equivariance to the Euclidean group. One option is to use a message-passing neural network as a backbone and compute/update messages equivariantly via convolutions. This approach generalizes well to point clouds and graphs and offers the high expressivity of graph neural networks. Equivariant convolutional operators can be further categorized as regular [3, 9, 14, 26] or steerable group convolutions [10, 11, 52]. The latter recently proved to be especially suitable for incorporating physical and geometric quantities into a model [5]. The key idea behind Steerable CNNs is using standard convolution - which guarantees translation equivariance - with $G$-steerable kernels that ensure commutativity with the transformations of another group $G$, such as rotations. The commutation requirement imposes a constraint onto the kernel space that must be solved analytically for each group $G$. This, in turn, does not allow generalizing a convolution operator tailored to a specific group to other symmetry transformations. In

---

[*]Work done while at Helmholtz-Zentrum Dresden-Rossendorf.
[†]Qualcomm AI Research is an initiative of Qualcomm Technologies, Inc.

the case of the Euclidean group, Cesa *et al.*[7] proposed a generally applicable way of parameterizing steerable convolutions for sub-groups of $E(n)$. The method relies on adapting a pre-defined kernel basis explicitly developed for the group $E(n)$ to an arbitrary sub-group by using *group restriction*.

However, because only a finite basis can be chosen, a basis tailored for $E(n)$ can be sub-optimal in terms of expressiveness for its sub-groups; see Section 3.6 more details. Hence, we propose an alternative way of building steerable convolutions based on implicit neural kernels, i.e. convolutional kernels implemented as continuous functions parameterized by MLPs [38, 39]. We demonstrate how $G$-steerable convolutions with implicit neural kernels can be implemented from scratch for any sub-group $G$ of the orthogonal group $O(n)$. The method allows us to ultimately minimize requirements to implement equivariance to new groups; see Section 3.3. The flexibility of neural functions also permits the injection of geometric and physical quantities in point convolutions, increasing their expressiveness [5]; see Section 3.2. We validate our framework on synthetic N-body simulation problem, point-cloud data (ModelNet-40 [56]) and molecular data (QM9 [55]) and demonstrate key bene-

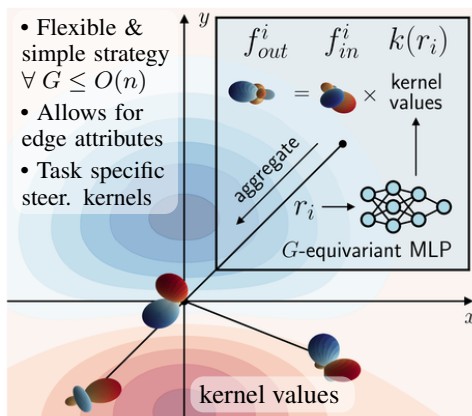

Figure 1: Illustration of the proposed approach: computing the response of an implicit kernel $k$ (background) of a steerable point convolution for the node $i$ (upper right corner) of a graph with steerable features (visualized as spherical functions).

fits of our approach such as flexibility and generalizability. Besides, we demonstrate that implicit kernels allow Steerable CNNs to achieve performance competitive with state-of-the-art models and surpass them with the correct choice of the task-specific symmetry group.

## 2  Background: Steerable Convolutions

In this work, we propose a general solution to easily build Steerable CNNs equivariant to translations *and* any compact group $G$[3]. In Section 2.1, we provide some necessary prerequisites from group theory and group representation theory [43]. Then, we review the framework of Steerable CNNs and discuss the constraint it induces on the convolutional kernels in Section 2.2.

### 2.1  Groups, Representations and Equivariance

**Definition 1** (Group action). *An action of a group $G$ on a set $\mathcal{X}$ is a mapping $(g, x) \mapsto g.x$ associating a group element $g \in G$ and a point $x \in \mathcal{X}$ with some other point on $\mathcal{X}$ such that the following holds:*

$$g.(h.x) = (gh).x \qquad \forall g, h \in G, x \in \mathcal{X}$$

**Definition 2** (Group representation). *A linear representation $\rho$ of a group $G$ is a map $\rho : G \to \mathbb{R}^{d \times d}$ that assigns an invertible matrix $\rho(g) \; \forall g \in G$ and satisfies the following condition*

$$\rho(gh) = \rho(g)\rho(h) \; \forall g, h \in G.$$

A group representation $\rho(g) : V \to V$ furthermore can be seen as a linear action of $G$ on a vector space $V$. Additionally, if two (or more) vectors $v_1 \in \mathbb{R}^{d_1}$ and $v_2 \in \mathbb{R}^{d_2}$ belong to vector spaces transforming under representations $\rho_1$ and $\rho_2$, their concatenation $v_1 \oplus v_2 \in \mathbb{R}^{d_1 + d_2}$ transforms under the *direct sum* representation $\rho_1 \oplus \rho_2$. $(\rho_1 \oplus \rho_2)(g)$ is a $d_1 + d_2$ dimensional block-diagonal matrix containing $\rho_1(g)$ and $\rho_2(g)$ in its diagonal blocks.

There are three types of group representations that are important for the definition of our method:

- **Trivial representation:** all elements of $G$ act on $V$ as the identity mapping of $V$, i.e. $\rho(g) = I$.
- **Standard representation:** the group $O(n)$ of all orthogonal $n \times n$ matrices has a natural action on $V = \mathbb{R}^n$; similarly, if $G$ is a subgroup of $O(n)$, elements of $G$ can act on $V = \mathbb{R}^n$ via the inclusion mapping, i.e. $\rho_{st}(g) = g \in \mathbb{R}^{n \times n}$.

---

[3]We provide the definition for a group in Appendix A.1.

- **Irreducible representations** is a collection of generally known representations that can be used as a building block for larger representations via *direct sum*. As argued in [49], Steerable CNNs can be parameterized without loss of generality solely in terms of irreducible representations.

**Definition 3** (Equivariance). *Let us have two spaces $\mathcal{X}, \mathcal{Y}$ endowed with a symmetry group $G$, i.e. with an action defined on them. A function $\phi : \mathcal{X} \to \mathcal{Y}$ is called $G$-equivariant, if it commutes with the action of $G$ on the two spaces, i.e. $\phi(g.x) = g.\phi(x)$ for all $g \in G$, $x \in X$.*

As we have discussed, the layers of conventional CNNs are translation equivariant by design; however, they do not commute with transformations of other groups, such as rotations and reflections.

## 2.2 Steerable CNNs

Steerable CNNs provide a more general framework that allows building convolutions that are equivariant to a group of *isometries* of $\mathbb{R}^n$, i.e. $(\mathbb{R}^n, +) \rtimes G \le E(n)$. Those groups are decomposable as a semi-direct[4] product of the translations group $(\mathbb{R}^n, +)$ and an origin-preserving compact[5] group $G \le O(n)$, where $O(n)$ is the group of $n$-dimensional rotations and reflections. As translation equivariance is guaranteed [52] by the convolution operator itself, one only has to ensure equivariance to $G$. See [51] for a more in-depth description of Steerable CNNs.

The feature spaces of Steerable CNNs are described as collections of *feature fields*. A feature field of type $\rho$ is a feature map $f : \mathbb{R}^n \to \mathbb{R}^d$ endowed with a *group representation* $\rho : G \to \mathbb{R}^{d \times d}$ that defines how an element $g \in G$ transforms the feature:

$$[g.f](x) := \rho(g)f(g^{-1}.x) \tag{1}$$

Furthermore, each convolutional layer is a map between feature fields. For the map to be equivariant, it must preserve the transformation laws of its input and output feature fields. In practice, it means that the following constraint onto the space of convolution kernels must be applied [52]:

$$k(g.x) = \rho_{out}(g)k(x)\rho_{in}(g)^{-1} \qquad \forall g \in G, x \in \mathbb{R}^n \tag{2}$$

where $k : \mathbb{R}^n \to \mathbb{R}^{d_{out} \times d_{in}}$, and $\rho_{in} : G \to \mathbb{R}^{d_{in} \times d_{in}}$, $\rho_{out} : G \to \mathbb{R}^{d_{out} \times d_{out}}$ are respective representations of input and output feature fields.

To parameterize the kernel $k$, the constraint in equation 2 needs to be solved *analytically* for each specific group $G$ of interest. This renders a general solution challenging to obtain and limits the applicability of steerable convolutions.

# 3 Implicit neural kernels

Instead of deriving a steerable kernel basis for each particular group $G$, we propose parameterizing the kernel $k : \mathbb{R}^n \to \mathbb{R}^{d_{out} \times d_{in}}$ with an MLP satisfying the constraint in Eq. 2. The approach only requires the $G$-equivariance of the MLP and suggests a flexible framework of implicit steerable convolutions that generalizes to arbitrary groups $G \le O(n)$. We argue about the minimal requirements of this approach in Section 3.3.

We first define the kernel as an equivariant map between vector spaces that we model with an MLP (see Section 3.1). Then, we demonstrate that $G$-equivariance of an MLP is a sufficient condition for building the implicit representation of steerable kernels for *compact* groups. We indicate that the flexibility of neural representation allows expanding the input of a steerable kernel in Section 3.2. Next, we describe how a $G$-equivariant MLP can be implemented in section 3.3. Later, we describe how one can implement $G$-steerable point convolution in the form of equivariant message passing [5, 40] in Section 3.4 and its generalization to dense convolution in Section 3.5. Finally, we compare our method with the solution strategy proposed in [7] in Section 3.6.

---

[4]See Appendix A.1 for the definition of a semi-direct product.
[5]To remain in the scope of the manuscript, we abstain from the mathematical definition of compact groups, which requires introducing topological groups. One can find more information about compact groups in [26].

## 3.1 Kernel vectorization and equivariance

Our goal is to implement the kernel $k : \mathbb{R}^n \to \mathbb{R}^{d_{out} \times d_{in}}$ of a $G$-steerable convolution that maps between spaces of feature fields with representations $\rho_{in}$ and $\rho_{out}$. The kernel itself is a function whose input in $\mathbb{R}^n$ transforms under the *standard representation* $\rho_{st}$ (as defined in Section 2.1) and which we will model with an MLP. Since MLPs typically output vectors, it is convenient to *vectorize* the $d_{out} \times d_{in}$ output of the kernel. We denote the column-wise vectorization of a matrix $M \in \mathbb{R}^{d_1 \times d_2}$ as $vec(M) \in \mathbb{R}^{d_1 d_2}$. Henceforth, we will consider kernel's vector form $vec(k(\cdot)) : \mathbb{R}^n \to \mathbb{R}^{d_{out} d_{in}}$.

Let $\otimes$ denote the *Kronecker product* between two matrices. Then, $\rho_{\otimes}(g) := \rho_{in}(g) \otimes \rho_{out}(g)$ is also a representation[6] of $G$. We suggest an implicit representation of the vectorized kernel $k$ using an $G$-equivariant MLP $\phi : \mathbb{R}^n \to \mathbb{R}^{d_{out} d_{in}}$ based on the following lemma (see A.2 for the proof):

**Lemma 1.** *If a kernel $k$ is parameterized by a $G$-equivariant MLP $\phi$ with input representation $\rho_{st}$ and output representation $\rho_{\otimes} := \rho_{in} \otimes \rho_{out}$ , i.e. $vec(k)(x) := \phi(x)$, then the kernel satisfies the equivariance constraint in Equation 2 for a compact group $G$.*

In other words, $G$-equivariance of MLP is a sufficient condition for $G$-equivariance of the convolutional layer whose kernel it parameterizes. Using implicit kernels also has a very favourable property - it allows arbitrary steerable features as its input, which we discuss in the following section.

## 3.2 Expanding the input

Note that in the case of standard steerable convolutions, the input $x \in \mathbb{R}^n$ of a kernel is usually only the difference between the spatial position of two points. However, there is no requirement that would disallow the expansion of the input space except for practical reasons. Hence, here we augment steerable kernels with an additional feature vector $z \in \mathbb{R}^{d_z}$. This formulation allows us to incorporate relevant information in convolutional layers, such as physical and geometric features. For example, when performing convolutions on molecular graphs, $z$ can encode the input and output atoms' types and yield different responses for different atoms. If introducing additional arguments into a kernel, the steerability constraint in equation 2 should be adapted to account for transformations of $z$:

$$k(g.x, \rho_z(g)z) = \rho_{out}(g)k(x,z)\rho_{in}(g)^{-1} \tag{3}$$

which must hold for all $g \in G, x \in \mathbb{R}^n, z \in \mathbb{R}^{d_z}$, where $\rho_z : G \to \mathbb{R}^{d_z \times d_z}$ is the representation of $G$ acting on the additional features.

Again, analytically solving the constraint 3 to find a kernel basis for arbitrary $\rho_z$ is generally unfeasible. Note also that the solution strategy proposed in [7] requires a basis for functions over $\mathbb{R}^{n+d_z}$, whose size tends to grow exponentially with $d_z$ and, therefore, is not suitable. Alternatively, we can now use the flexibility of neural representation and introduce additional features into a kernel at no cost.

## 3.3 Implementing a $G$-equivariant MLP

We are now interested in how to build an MLP that is equivariant to the transformations of the group $G$, i.e. a sequence of equivariant linear layers alternated with equivariant non-linearities [15, 37, 44]. It is important to say that our approach does not rely on a specific implementation of $G$-MLPs, and any algorithm of preference might be used (e.g. [15] or enforcing via an additional loss term).

The approach we employed in our experiments is described below. Since the irreducible representations of $G$ are typically known[7], one can always rely on the following properties: *1)* any representation $\rho$ of a compact group $G$ can be decomposed as a direct sum of irreducible representations $\rho(g) = Q^T \left( \bigoplus_{i \in I} \psi_i(g) \right) Q$ with a change of basis $Q$[8] and *2) Schur's Lemma*, which states that there exist equivariant linear maps only between the irreducible representation of the same kind[9]. Hence, one can apply the right change of basis to the input and output of a linear layer and then learn only maps between input and output channels associated with the same irreducible representations. In the context of implicit kernels, the tensor product representation $\rho_{in} \otimes \rho_{out}$ in the last layer is

---

[6]This representation is formally known as the *tensor product* of the two representations.

[7]This is a reasonable assumption since irreducible representations are often used to construct other representations used in Steerable CNNs. If not, there exist numerical methods to discover them from the group algebra.

[8][7] describes and implements a numerical method for this.

[9]Typically, these maps only include scalar multiples of the identity matrix.

decomposed by a matrix containing the Clebsch-Gordan coefficients, which often appears in the analytical solutions of the kernel constraint in the related works. Note that the non-linearity $\sigma$ used in the Steerable CNNs are $G$-equivariant and, therefore, can be used for the MLP as well.

## 3.4 $G$-steerable point convolutions

Point convolutions operate on point clouds - sets of $N$ points endowed with spatial information $X = \{x_i\}_{i=0}^{N-1} \in \mathbb{R}^{n \times N}$. A point cloud thus provides a natural discretization of the data domain, which renders a convolution operator as follows:

$$f_{out}(x_i) = (k * f_{in})(x_i) = \sum_{0 \leq j \leq N-1} k(x_i - x_j) f_{in}(x_j) \tag{4}$$

To reduce the computational cost for large objects, one can induce connectivity onto $x \in X$ and represent a point cloud as a graph $\mathcal{G} = (\mathcal{V}, \mathcal{E})$ with nodes $v_i \in \mathcal{V}$ and edges $e_{ij} \in \mathcal{E} \subseteq \mathcal{V} \times \mathcal{V}$, where each node $v_i$ has a spatial location $x_i$, node features $z_i$ and a corresponding feature map $f_{in}(x_i)$ (see Figure 1). Additionally, each edge $e_{ij}$ can have an attribute vector $z_{ij}$ assigned to it (as in Section 3.2). This allows for a learnable message-passing point convolution whose computational cost scales linearly with the number of edges:

$$f_{out}(x_i) = (k * f_{in})(x_i) = \sum_{j \in \mathcal{N}(i)} k(x_i - x_j, z_i, z_j, z_{ij}) f_{in}(x_j) \tag{5}$$

where $\mathcal{N}(i) = \{j : (v_i, v_j) \in \mathcal{E}\}$ and the kernel $k(\cdot)$ is parameterized by a $G$-equivariant MLP.

## 3.5 Extension to $G$-steerable CNNs

Straightforwardly, the proposed method can be extended to dense convolutions. In such case, the kernel is defined as $k : \mathbb{R}^n \to \mathbb{R}^{c_{out} \times c_{in} \times K^n}$ - that is, a continuous function that returns a collection of $K \times K \times ...$ kernels given a relative position (and, optionally, arbitrary steerable features). In this case, the vectorized kernel is parameterized in exactly the same way as described above but is estimated at the center of each pixel.

## 3.6 Comparison with the analytical solution

A general basis for any $G$-steerable kernel is described in [7]. Essentially, it relies on two ingredients: *i)* a (pre-defined) finite $G$-steerable basis [17] for *scalar* filters and *ii)* a learnable equivariant linear map. Let us look at those in detail. Firstly, a finite $G$-steerable basis is essentially a collection of $B$ orthogonal functions, i.e. $Y : \mathbb{R}^n \to \mathbb{R}^B$, with the following equivariant property: $Y(g.x) = \rho_Y(g) Y(x)$, for some representation $\rho_Y$ of $G$. The linear map in *ii)*, then, is a general equivariant linear layer, whose input and output transform as $\rho_Y$ and $\rho_{in} \otimes \rho_{out}$.

In practice, it means that one has to provide a pre-defined basis $Y$ for a group of interest $G$. Since the design might not be straightforward, [7] suggest a way to *reuse* an already derived $O(n)$ steerable basis for any subgroup $G \subset O(n)$. While general, such a solution can be sub-optimal. For example, if $n = 3$, an $O(3)$-steerable basis has local support inside a sphere, which is suitable for the group of 3D rotations $SO(3)$ but not ideal for cylindrical symmetries, i.e. when $G$ is the group $SO(2)$ of planar rotations around the Z axis.

In comparison, the solution proposed in Section 3.1 replaces the pre-defined basis $Y$ with a learnable $G$-MLP. It allows us to learn $G$-specific kernels without relying on a pre-derived basis for a larger group, which in turn means that we can theoretically obtain $G$-optimal kernel basis via learning (see A.3 for further details). Furthermore, the kernels defined in Section 3.1 can now be conditioned on arbitrary steerable features, which makes them more task-specific and expressive. In the context of a general basis, one can interpret the last linear layer of an implicit kernel as the map in *ii)*, and the activations before this layer as a learnable version of the basis $Y$ in *i)*.

# 4 Related works

**Group convolutions.** Multiple architectures were proposed to achieve equivariance to a certain symmetry group. It has been proven [26] that the convolutional structure is a sufficient condition for

building a model that is equivariant to translations and actions of a compact group. One can separate group convolutions into two classes depending on the space on which the convolution operates: *regular* [3, 4, 9, 14, 26] and *steerable* [7, 10, 11, 46, 49, 52, 54] group convolutions. In the first case, the input signal is represented in terms of scalar fields on a group $G$, and the convolution relies on a discretization of the group space. Steerable convolutions are a class of $G$-equivariant convolutions that operate on feature fields over homogeneous spaces and achieve equivariance via constraining the kernel space. They further avoid the discretization of the group space and can reduce the equivariance error in the case of continuous groups.

$G$**-steerable kernels.** In the case of rigid body motions $G = SO(3)$, the solution of the equivariance constraint is given by spherical harmonics modulated by an arbitrary continuous radial function, which was analytically obtained by Weiler *et al.*[52]. Lang and Weiler [27] then applied the Wigner-Eckart theorem to parametrize $G$-steerable kernel spaces over orbits of a compact $G$. The approach was later generalized by Cesa *et al.*[7], who proposed a solution for any compact sub-group of $O(3)$ based on group restriction. Using this approach, one can obtain a kernel basis for a group $G \leq H$ if the basis for $H$ is known. Despite its generalizability, the method still requires a pre-defined basis for the group $H$ that is further adapted to $G$. The resulting solution is not guaranteed to be optimal for $G$; see Section 3.6. We note that the practical value of steerable kernels is also high as they can be used for convolution over arbitrary manifolds in the framework of Gauge CNNs [8, 21, 50].

**Implicit kernels.** Using the implicit representation of convolution kernels for regular CNNs is not novel. It was used, for example, to model long-range dependencies in sequential data [39] or for signal representation [45]. Romero *et al.*[39] demonstrated that such parametrization allows building shallower networks, thus requiring less computational resources to capture global information about the system. Continuous kernels were recently used to build an architecture [25] for processing data of arbitrary resolution, dimensionality and length, yet equivariant solutions are scarce [47]. Finzi *et al.*[14] proposed parametrizing convolutions on Lie groups as continuous scalar functions in the group space. The method relies on discretizing a continuous group, which might lead to undesirable stochasticity of the model's output. Instead, we use Steerable CNNs that define the kernel as a function on a homogeneous space. While the discretization of this space is still required, in most cases, it is naturally given by the data itself, e.g. for point clouds; hence, no sampling error arises. It is also important to mention the key difference between implicit kernels and applying $G$-MLPs [15] directly - the latter is incapable of processing image/volumetric data as convolutions do. Henceforth, we focus on CNNs with consideration for potential extensions to various data modalities.

**Equivariant point convolutions.** A particular momentum has been gained by point convolutions in the form of equivariant message-passing [5, 40, 41, 46] specifically for problems where symmetry provides a strong inductive bias such as molecular modelling [1] or physical simulations [18]. Thomas *et al.*[46] pioneered $SE(3)$-equivariant steerable convolutions whose kernels are based on spherical harmonics modulated by a radial function. The approach was further generalized by Batzner *et al.*[2], who uses an MLP conditioned on the relative location to parameterize the radial function, although the basis of spherical harmonics is preserved. Brandstetter *et al.*[5] demonstrated that introducing geometric and physical information into an equivariant message-passing model improves the expressivity on various tasks, which we also observe in this work. Note that, for $G = SO(3)$ or $O(3)$ and without additional edge features $z$, our MLP can only learn a function of the radius and, therefore, is equivalent to the models proposed in [2].

## 5 Experiments

In this section, we implement Steerable CNNs with implicit kernels and apply them to various tasks[10]. First, we indicate the importance of correctly choosing the symmetry group on a synthetic N-body simulation problem where an external axial force breaks the rotational symmetry (see Section 5.2). Then, we prove the generalizability of the proposed approach as well as the gain in performance compared to the method proposed in [7] on ModelNet-40 (see Section 5.3). Afterwards, we show that one can introduce additional physical information into a kernel and significantly improve the performance of steerable convolutions on molecular data (see Section 5.4). Code and data to reproduce all experiments are available on GitHub.

---

[10]All datasets were downloaded and evaluated by Maksim Zhdanov (University of Amsterdam).

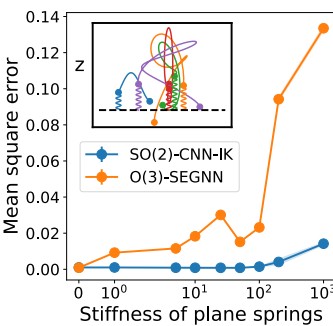

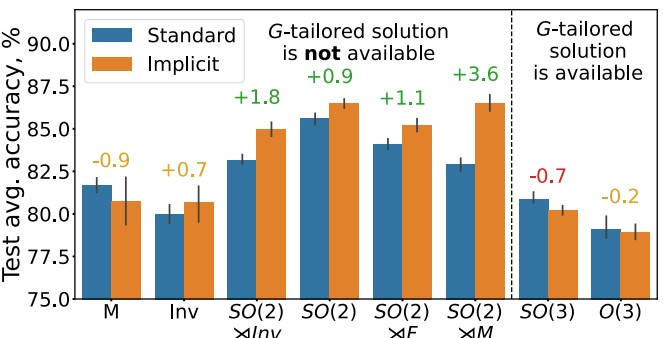

Figure 2: Final position estimation in the N-body system experiment, where particles are connected to one another and to the XY-plane by springs (see left upper corner). Our model with correct axial symmetry $SO(2)$, significantly outperforms the state-of-the-art model SEGNN [5], which is $O(3)$-equivariant, as the relative contribution of the plane string increases.

Figure 3: Performance comparison of Steerable CNNs on the rotated ModelNet-40 dataset for different $G$. Bars show mean average accuracy with error bars indicating standard deviation, both computed from 5 runs. The numbers above bars denote the performance gain of implicit kernels (orange) over [7] (blue). Statistically significant ($p < 0.05$) positive differences are green, negative ones are red, and insignificant ones are yellow. $SO(2)$ contains planar rotations around the Z axis, $M$ and $Inv$ contain mirroring along the $X$ axis and origin respectively, while $F$ contains rotations by $\pi$ around the $X$ axis. $O(2) \cong SO(2) \rtimes M$ achieves the best accuracy as it best represents the real symmetries of the data.

### 5.1 Implementation

**Implicit kernels.** To parameterize implicit kernels, we employ linear layers (see Section 3.3) followed by quotient ELU non-linearities [7]. The last layer generates a steerable vector, which we reshape to yield a final convolutional kernel. The $G$-MLP takes as input a steerable vector obtained via direct sum (concatenation) of batch-normalized harmonic polynomial representation of the relative location $x$, edge features $z_{ij}$ and input node features $z_i$. See details on pre-processing and task-specific inputs in Appendix B.1.

**Steerable convolutions.** We employ steerable point convolutional layers as described in Equation 5. For each task, we tune hyperparameters of all models on validation data: the number of layers, the number of channels in each layer, the depth and the width of implicit kernels, and the number of training epochs. For all the experiments, except the one reported in Table 2, we yield a number of parameters similar to the baselines with a deviation of less than 10%. For ModelNet-40 experiments, our architecture is partially inspired by the model introduced in [34], which uses gradual downsampling of a point cloud[11]. For N-body and QM9 experiments, we add residual connections to each steerable layer to learn higher-frequency features [20]. For QM9 and ModelNet-40 experiments, the last steerable layer returns a vector with invariant features for each node, to which we apply global pooling to obtain a global representation of an object, which is further passed to a classification MLP. In the case of N-body experiments, we output the standard representation corresponding to the particles' coordinates. Details about optimization and model implementation for each task can be found in Appendix B.2.

### 5.2 The relevance of smaller $G < O(n)$: N-body simulation

**Dataset.** We conduct experiments using the N-body system [23], where particles are connected through springs, and their interaction is governed by Hooke's law. Similar to previous studies [5, 40], we modify the original trajectory prediction task to calculate the position of each particle in a 3-dimensional space, given its initial position and velocity. We attach each particle to the XY plane using strings with random equilibrium lengths and pre-defined stiffness, which can slide freely over the plane (see Figure 2, upper left corner). This additional force term breaks the rotational symmetry of the system into only azimuthal symmetry; this system resembles a simplified version

---

[11]Poulenard *et al.*[34] use kd-tree pooling to compute coarser point clouds, while we only use random sampling of points. Note also that the spherical quotient ELU we used is similar in spirit to the proposed functional non-linearity described there, yet it does not employ a deep MLP.

of the problem of molecule binding to the surface of a larger molecule. We choose the model's hyperparameters to have a similar parameter budget to the SEGNN model [5], which has shown state-of-the-art performance on a similar task (we also compare against a non-equivariant baseline; see Table A2). For all models, we use the highest frequency of hidden representations in both $G$-MLP and Steerable CNNs equal to 1. We use the velocity of a particle and the equilibrium length of the attached XY spring as input; the model's output transforms under the standard representation. We train a separate model for each value of plane strings' stiffness (3000 training points) to measure the impact of symmetry breaks on performance.

**Results.** As can be seen in Fig. 2, a Steerable CNN with azimuthal symmetry $SO(2)$ significantly outperforms SEGNN, which is equivariant to a larger group $O(3)$. Since we introduced a force term that now breaks the rotational symmetry, SEGNN struggles to learn it. Furthermore, while in the default setting (plane strings' stiffness is 0), models achieve roughly the same performance, the absolute difference grows exponentially once the plane strings are introduced.

The synthetic problem is meant to show the importance of choosing the correct symmetry when designing a model. While the Euclidean group E(3) is often enough for N body systems in a vacuum, it is important to be careful when an external influence or underlying structure can disrupt global symmetries and negatively impact a model with a larger symmetry group. This can be relevant in scenarios like molecular docking simulations [12], where the system can align with the larger molecule, or material science [22], where the arrangement of atoms and crystal lattice structures yield a discrete group of symmetries smaller than $O(n)$.

## 5.3 Generalizability of implicit kernels: ModelNet-40

**Dataset.** The ModelNet-40 [56] dataset contains 12311 CAD models from the 40 categories of furniture with the orientation of each object aligned. The task is to predict the category of an object based on its point cloud model. 2468 models are reserved for the test partition. From the remaining objects, we take 80% for training and 20% for validation. We augment each partition with random rotations around the Z-axis. We induce connectivity on point clouds with a k-nearest neighbour search with $k = 10$ at each model layer and use normals as input node features.

**Results.** In this section, we demonstrate how one can build a Steerable CNN that is equivariant to an arbitrary subgroup of the Euclidean group $G \rtimes (\mathbb{R}^3, +) \leq E(3)$. We compare the performance of implicit kernels with the standard steerable kernels obtained by group restriction [7] and keep the number of parameters similar. The results are shown in Figure 3. Implicit kernels achieve significant improvement in accuracy on test data for the majority of groups. The only statistically significant negative difference is presented for $G = SO(3)$, for which a tailored and hence optimal kernel basis is already available. When a custom solution is unknown, implicit kernels often significantly outperform the previously proposed method. There-

Table 1: Overall accuracy (OA) on ModelNet-40. Group equivariant methods are denoted by $*$.

| Method | OA, % |
|---|---|
| Spherical-CNN [13] $*$ | 88.9 |
| SE(3)-ESN [16] $*$ | 89.1 |
| TFN[mlp] P [34] $*$ | 89.4 |
| PointNet++ [35] | 91.8 |
| SFCNN [36] | 92.3 |
| PointMLP [31] | 94.5 |
| Ours | $89.3 \pm 0.06$ |

fore, they pose an efficient toolkit for building a kernel basis for an arbitrary subgroup of $E(3)$. We report the result of the best-performing model in Table 1. Although we do not outperform more task-specific and involved approaches such as PointMLP [31], our model is on par with other group equivariant models. We emphasize that our model is a simple stack of convolutional layers and hypothesize that a smarter downsampling strategy, residual connections and hierarchical representations would significantly improve the overall performance of our model, which we leave for further research.

## 5.4 Flexibility of implicit kernels: QM9

**Dataset.** The QM9 dataset [55] is a public dataset consisting of about 130k molecules with up to 29 atoms per molecule. Each molecule is represented by a graph with nodes denoting atoms and edges indicating covalent bonds. Each node is assigned a feature vector consisting of one-hot encoding of the type of the respective atom (H, C, N, O, F) and its spatial information corresponding to a low energy conformation. Additionally, each molecule is described by 19 properties from which we select

Table 2: Mean Absolute Error (MAE) between model predictions and ground truth for the molecular property prediction on the QM9 dataset. Linear steerable convolutions are denoted by $*$. L stands for the number of layers, and W stands for the number of channels in each layer.

| Task | $\alpha$ | $\Delta\varepsilon$ | $\varepsilon_{HOMO}$ | $\varepsilon_{LUMO}$ | $\mu$ | $C_\nu$ | G | H | $R^2$ | U | $U_0$ | ZPVE |
|------|------|------|------|------|------|------|------|------|------|------|------|------|
| Units | bohr$^3$ | meV | meV | meV | D | cal/mol K | meV | meV | bohr$^3$ | meV | meV | meV |
| NMP[19] | .092 | 69 | 43 | 38 | .030 | .040 | 19 | 17 | 0.180 | 20 | 20 | 1.50 |
| SchNet[41] | .235 | 63 | 41 | 34 | .033 | .033 | 14 | 14 | 0.073 | 19 | 14 | 1.70 |
| SE(3)-Tr.[18] | .142 | 53 | 35 | 33 | .051 | .054 | - | - | - | - | - | - |
| DimeNet++[24] | .043 | 32 | 24 | 19 | .029 | .023 | 7 | 6 | 0.331 | 6 | 6 | 1.21 |
| SphereNet[29] | .046 | 32 | 23 | 18 | .026 | .021 | 8 | 6 | 0.292 | 7 | 6 | 1.12 |
| PaiNN[42] | .045 | 45 | 27 | 20 | .012 | .024 | 7 | 6 | 0.066 | 5 | 5 | 1.28 |
| EGNN[40] | .071 | 48 | 29 | 25 | .029 | .031 | 12 | 12 | 0.106 | 12 | 12 | 1.55 |
| SEGNN[5] | .060 | 42 | 24 | 21 | .023 | .031 | 15 | 16 | 0.660 | 13 | 15 | 1.62 |
| TFN[46]$*$ | .223 | 58 | 40 | 38 | .064 | .101 | - | - | - | - | - | - |
| Cormorant[1]$*$ | .085 | 61 | 34 | 38 | .038 | .026 | 20 | 21 | 0.961 | 21 | 22 | 2.02 |
| L1Net[32]$*$ | .088 | 68 | 46 | 35 | .043 | .031 | 14 | 14 | 0.354 | 14 | 13 | 1.56 |
| LieConv [14]$*$ | .084 | 49 | 30 | 25 | .032 | .038 | 22 | 24 | 0.800 | 19 | 19 | 2.28 |
| Ours (W=24, L=15) | .078 | 45.3 | 24.1 | 22.3 | .033 | .032 | 21.1 | 19.6 | 0.809 | 19.7 | 19.5 | 2.08 |
| Ours (W=16, L=30) | .077 | 43.5 | 22.8 | 22.7 | .029 | .032 | 19.9 | 21.5 | 0.851 | 22.5 | 22.3 | 1.99 |

12 commonly taken in literature [5] for regression tasks. Different from common practice [5], we perform convolution on molecular graphs, i.e. with connectivity pre-defined by molecular structure instead of inducing it. The design choice is motivated by the presence of edge features, which we include in an implicit kernel to display the flexibility of neural representation.

**Results.** We first demonstrate how one can use the flexibility of neural representation to introduce additional features of choice into a steerable convolutional layer. As each pair of atoms connected by a covalent bond is assigned a one-hot encoding of the bond type $z_{ij}$, we use it as a condition for $O(3)$-equivariant implicit kernels. Additionally, we follow Musaelian *et al.*[33] and embed one-hot encoding of the center and neighbour atom types $z_i$ and $z_j$ into the MLP. We include each property one by one and indicate corresponding performance gain in Figure 4. First, we observed a sharp improvement by switching from standard steerable kernels to implicit ones. We attribute it to the higher expressivity of implicit kernels and their ability to learn more complex interactions. Furthermore, injecting edge attributes into the kernel computation reduced MAE even further. Introducing both atom types and an edge type significantly influenced the per-

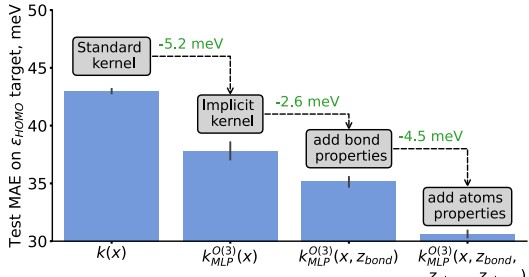

Figure 4: Using implicit kernels $k^G_{MLP}$ and injecting it with bond and atoms properties significantly improves the performance of Steerable CNNs on the QM9 dataset (Mean Absolute Error on the $\varepsilon_{HOMO}$ regression problem). Bars denote mean average accuracy on the test dataset with error bars corresponding to standard deviation; both computed on 5 runs. The kernel is $O(3)$-equivariant rendering the final architecture $E(3)$-equivariant.

formance, corresponding to the model learning how to process each specific combination differently, thus adding to its expressivity. It is not surprising since a similar result was obtained by Brandstetter *et al.*[5], who used non-linear message aggregation conditioned on physical information, yielding state-of-the-art performance.

**Scaling the model up.** Table 2 shows the results of an $E(3)$-equivariant Steerable CNN with implicit kernels on the QM9 dataset. Both models we reported here have approx. $2 \cdot 10^6$ parameters and only differ in length and width of the convolutional layers. For non-energy regression tasks ($\alpha$, $\Delta\varepsilon$, $\varepsilon_{HOMO}$, $\varepsilon_{LUMO}$, $\mu$ and $C_\nu$), we obtain results that are on par with the best-performing message-passing based approaches (we do not compare against transformers). We also indicate that Steerable CNNs with implicit kernels significantly outperform steerable linear convolutions (TFN, LieConv, L1Net) on most tasks. This is consistent with the observation of Brandstetter *et al.*[5], who pointed out that non-linear convolutions generally perform better than linear ones. For the remaining energy variables (G, H, U, $U_0$, ZPVE) and $R^2$, our model significantly falls behind the task-specific benchmark approaches. We theorize that it can be attributed to two factors. First, steerable convolutions generally do not perform well on these tasks compared to problem-tailored

frameworks (PaiNN [42], DimeNet++ [24], SphereNet [29]), also in the non-linear case (e.g. SEGNN [5]). Second, we hypothesize that molecular connectivity does not produce a sufficient number of atom-atom interactions, which is crucial for the performance of a message-passing-based model [5]. However, as the goal of the section was to demonstrate the flexibility of implicit kernels that can be conditioned on features of graph edges, we leave developing more involved architectures (e.g. with induced connectivity) for further work.

## 6 Conclusion

We propose a novel approach for implementing convolutional kernels of Steerable CNNs, allowing for the use of smaller groups and easy integration of additional features under the same framework with minor changes. To avoid analytically solving the group $G$-specific equivariance constraint, we use a $G$-equivariant MLP to parameterize a $G$-steerable kernel basis. We theoretically prove that MLP equivariance is sufficient for building an equivariant steerable convolutional layer. Our implicit representation outperforms a previous general method, offering a way to implement equivariance to various groups for which a custom kernel basis has not been developed. The N-body experiment suggests that this method will be particularly applicable in scenarios where rotational symmetry is disturbed, such as in material science or computational chemistry. The critical force term, which violated rotational symmetry, was effectively captured by our model, while the state-of-the-art model struggled with it. Additionally, our flexible neural representation enables the introduction of arbitrary features into a convolutional kernel, enhancing the expressivity of Steerable CNNs. We validate this advantage by applying Steerable CNNs to point cloud and molecular graph data and achieving competitive performance with state-of-the-art approaches. In conclusion, we present a simple yet efficient solution for constructing a general kernel basis equivariant to an arbitrary compact group.

## Limitations

Steerable CNNs generally have high computational complexity and higher memory requirements compared to traditional CNNs. Implicit neural kernels do not help to mitigate the issue, yet they provide additional control over the kernel complexity. We found that when the kernels are parameterized by a single linear layer, the run time slightly decreases compared to the analytical solution[12], while a relative performance gain remains. We suggest that more efficient ways to implement $G$-MLPs would significantly contribute to the acceleration of the method and leave it for further research. From the implementation point of view, the most troublesome was achieving initialization similar to the non-implicit convolutions. We expect the problem to manifest even stronger when dealing with dense convolutions. This, combined with the discretization error of matrix filters, might negatively affect the performance. We are, however, convinced that it is a matter of time before a robust way of initializing $G$-equivariant implicit kernels will be obtained.

## Acknowledgments and Disclosure of Funding

All the experiments were performed using the Hemera compute cluster of Helmholtz-Zentrum Dresden-Rossendorf and the IvI cluster of the University of Amsterdam. This research results from a collaboration initiated at the London Geometry and Machine Learning Summer School 2022 (LOGML). The authors thank Anna Mészáros, Chen Cai and Ahmad Hammoudeh for their help at the initial stage of the project. They also thank Rob Hesselink for his assistance with visualizations.

---

[12]as implemented in `escnn` [7]

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
