# A   Theoretical details

This section provides an additional mathematical background that might be useful for understanding Steerable CNNs (Section A.1). Besides, we write down the proof of the cornerstone Lemma 1, which allows the application of implicit kernels (Section A.2). A more comprehensive introduction to representation theory and Steerable CNNs can be found in [51]. Finally, we highlight the difference between our method and the one described in [7] in terms of prerequisites in Section A.3.

## A.1   Additional details and definitions on group theory

**Definition 4** (Group). *A group is an algebraic structure that consists of a set $G$ and a binary operator $\circ : G \times G \to G$ called the group product (denoted by juxtaposition for brevity $g \circ h = gh$) that satisfies the following axioms:*

- $\forall g, h \in G : gh \in G$;
- $\forall g \in G \, \exists! \, g^{-1} \in G : gg^{-1} = g^{-1}g = e$;
- $\exists! \, e \in G : eg = ge = g \, \forall g \in G$;
- $(gh)k = g(hk) \, \forall g, h, k \in G$.

**Example 1** (The Euclidean group $E(3)$). *The 3D Euclidean group $E(3)$ comprises three-dimensional translations, rotations, and reflections. These transformations are defined by a translation vector $x \in \mathbb{R}^3$ and an orthogonal transformation matrix $R \in O(3)$. The group product and inverse are defined as follows:*

- $g \cdot g' := (Rx' + x, RR')$;
- $g^{-1} := (R^{-1}x, R^{-1})$,

*where $g = (x, R)$ and $g' = (x', R')$ are elements of $E(3)$. These definitions satisfy the four group axioms, establishing E(3) as a group. The action of an element $g \in E(3)$ on a position vector $y$ is given by:*

$$g \cdot y := Ry + x,$$

*where $g = (x, R)$ is an element of $E(3)$ and $y \in \mathbb{R}^3$.*

**Definition 5** (Semi-direct product). *Let $N$ and $H$ be two groups, each with their own group product, which we denote with the same symbol $\cdot$ , and let $H$ act on $N$ by the action $\odot$. Then a (outer) semi-direct product $G = N \rtimes H$, called the semi-direct product of $H$ acting on $N$ , is a group whose set of elements is the Cartesian product $N \times H$, and which has group product and inverse:*

$$(n, h) \cdot (\hat{n}, \hat{h}) = (n \cdot (h \odot \hat{n}), h \cdot \hat{h})$$

$$(n, h)^{-1} = (h^{-1} \odot n^{-1}, h^{-1})$$

for all $n, \hat{n} \in N$ and $h, \hat{h} \in H$.

## A.2   Proof of the Lemma 1

**Lemma.** *If a kernel $k$ is parameterized by a $G$-equivariant MLP $\phi$ with input representation $\rho_{st}$ and output representation $\rho_{\otimes} := \rho_{in} \otimes \rho_{out}$ , i.e. $vec(k)(x) := \phi(x)$, then the kernel satisfies the equivariance constraint in Equation 2 for a compact group $G$.*

*Proof.* By construction, the equivariant MLP satisfies

$$\phi(\rho_{st}(g)x) = (\rho_{in}(g) \otimes \rho_{out}(g)) \, \phi(x) \quad \forall g \in G, x \in \mathbb{R}^n \tag{6}$$

We further use the substitution and $\phi \mapsto vec(k(\cdot))$ and find:

$$vec(k(g.x)) = (\rho_{in}(g) \otimes \rho_{out}(g))vec(k(x)) \tag{7}$$

Now, we make use of the following identity describing the vectorization of a product of multiple matrices, which is the property of the Kronecker product:

$$vec(ABC) = (C^T \otimes A) \, vec(B) \tag{8}$$

Hence, identity 8 allows us to re-write the previous equation as follows:

$$k(g.x) = \rho_{out}(g)k(x)\rho_{in}(g)^T \tag{9}$$

Since we assume $G$ to be compact, its representations can always be transformed to an orthogonal form for which $\rho(g)^T = \rho(g)^{-1}$ via a change of basis. Hence, we find the equivariance constraint defined in equation 2. $\qquad\square$

### A.3 Additional details on the comparison with [7]

To summarize the difference between our method and [7], we provide Table A1. There, we highlight the key ingredients required for the implementation of Steerable kernels for a group $G$ in both methods and estimate the "hardness" of obtaining each ingredient. As can be seen, the method described in [7] required a $G$-steerable basis for $L^2(\mathbb{R}^n)$, while implicit kernels do not. Instead, one only has to provide $G$-equivariant non-linearities, which we assume are available since those are the same non-linearities that will be used in the main model.

Table A1: Key ingredients required to build $G$-steerable kernels with baseline [7] (centre left) vs implicit kernels (centre right). The left column highlights the general prerequisites of Steerable CNNs, and the right column indicates the relative complexity of each ingredient.

| Requirement | Hardness |
|---|---|
| **Design of Steerable CNN architecture †** | |
| irreps $\hat{G}$ of $G$ | assumed |
| action of $G$ on $\mathbb{R}^n$ | assumed |
| $G$-equivariant non-linearities | assumed |
| **Solve constraint with [7]** | |
| CG coefficients for $G$ | numerical |
| intertwiners $E_G(V_\psi)$ for $\psi \in \hat{G}$ | numerical or analytical |
| irreps-decomposition of $\rho_{in}$ and $\rho_{out}$ | numerical |
| $G$-steerable basis for $L^2(\mathbb{R}^n)$ | handcrafted ad-hoc for each $G$ |
| **Implicit Kernel (Ours)** | |
| CG coefficients for $G$ | numerical |
| intertwiners $E_G(V_\psi)$ for $\psi \in \hat{G}$ | numerical or analytical |
| irreps-decomposition of $\rho_{in}$ and $\rho_{out}$ | numerical |
| $G$-equivariant non-linearities | available from † |

## B  Experimental details

The section aims to provide additional details on model implementation for each particular experiment in Section 5. First, we describe how we preprocess the input of implicit kernels (relative position $x_i - x_j$, node features $z_i$, $z_j$, and edge features $z_{ij}$), which holds for every model (Section B.1). Then, we report the architectural details for models used in every experiment (Section B.2).

### B.1  Preprocessing kernel's input

An implicit kernel receives as input the relative position $x_i - x_j$, node features $z_i$ and $z_j$, and edge features $z_{ij}$. The first argument is a set of 3-dimensional points transforming according to the standard representation $\rho_{st}$. We first compute its *homogeneous polynomial representation* in $\mathbb{R}^3$ up to order $L$ and then batch-normalize it separately for each irrep before passing it to the implicit kernel. The harmonic polynomial $Y_l(x)$ of order $l$ evaluated on a point $x \in \mathbb{R}^3$ is a $2l + 1$ dimensional vector transforming according to the Wigner-D matrix of frequency $l$ (and parity $l \mod 2$, when interpreted as an irrep of $O(3)$). The vector $Y_l(x)$ is computed by projecting $x^{\otimes l} \in \mathbb{R}^{3^l}$ on its only $2l + 1$ dimensional subspace transforming under the frequency $l$ Wigner-D matrix.[13] We keep $L = 3$ as

---

[13]If $D_l$ is the frequency $l$ Wigner-D matrix, $x$ transforms under $D_l$, which is isomorphic to the standard representation of $SO(3)$. Then, $x^{\otimes l}$ transforms under $D_1^{\otimes l}$, i.e. the tensor product of $l$ copies of $D_1$. The tensor

Table A2: Mean square error in the N-body system experiment vs stiffness of the strings from particles to the $XY$-plane. Stiffness practically indicates the degree of breaking the $SO(3)$ symmetry.

| Stiffness | 0 | 1 | 5 | 10 | 25 | 50 | 100 | 200 | 1000 |
|---|---|---|---|---|---|---|---|---|---|
| MPNN | 0.0022 | 0.0031 | 0.0068 | 0.0087 | 0.0030 | 0.0162 | 0.0560 | 0.0978 | 0.1065 |
| O(3)-SEGNN | 0.0009 | 0.0092 | 0.0117 | 0.0183 | 0.0291 | 0.0151 | 0.0229 | 0.0938 | 0.1313 |
| SO(2)-CNN-IK | 0.0010 | 0.0010 | 0.0009 | 0.0008 | 0.0008 | 0.0008 | 0.0014 | 0.0043 | 0.0162 |

we found the choice to be favourable for overall performance on validation data for ModelNet-40 and QM9 experiments. In the N-body experiment, $L = 1$ for a fair comparison with the baseline. The remaining arguments form a vector with a pre-defined representation, which we concatenate to the harmonic representation. We extensively compared different preprocessing techniques, and the batch normalization of polynomial representation improved the performance of implicit kernels the most. We attribute it to higher numerical stability as we discovered that the standard deviation of MLP's output is the lowest in the case. The output of the implicit representation is a vector which we multiply with a Gaussian radial shell $\phi(x) = exp(-0.5 \cdot ||x||_2^2/\sigma^2)$ where $\sigma$ is a learnable parameter. This is coherent with the kernel basis typically used in literature [53] - spherical harmonics modulated by a Gaussian radial shell.

## B.2    Model implementation

**N-body**    We used the reported configuration of the SEGNN model according to the official repository [5], which has around $10^4$ parameters. We only modified the input of the model such that it takes the equilibrium length of the attached XY spring instead of the product of charges as in the original formulation, which was a trivial representation as well. As a result, the input representation consisted of 2 standard representations (position and velocity) and a trivial representation (spring's equilibrium length). The training was performed precisely according to the official configuration. For the non-equivariant baseline, we substituted every equivariant MLP in SEGNN with its non-equivariant counterpart and adjusted the number of parameters to match the one of the original.

To form a dataset, 3000 trajectories were generated with random initial velocities and equilibrium lengths of XY-plane strings for training and 128 for validation and testing. $G$-equivariant MLPs had 3 layers with 16 hidden fields. We used an embedding linear layer followed by 4 steerable convolutions and a $G$-equivariant MLP with 2 hidden layers applied to each node separately. The hidden representation was kept the same across every part of the model and had 16 steerable vector fields transforming under the spherical quotient representation band-limited to maximum frequency 1. The total number of parameters was approximately equal to the one of the baseline. The model returned a coordinate vector transforming under the standard representation for each particle as output. We trained each model using a batch size of 128 for 200 epochs until reaching convergence. As in the case of SEGNN, we minimized the MSE loss. We used AdamW optimizer with an initial learning rate of $10^{-2}$. The learning rate was reduced by $0.5$ every 25 epochs. The training time, on average, was 5 min.

**ModelNet-40**    In the generalizability experiments described in Section 5.3, we maintained the configuration of the $G$-equivariant MLP as follows: two linear layers with 8 hidden fields and spherical quotient ELU with a maximum frequency of 2 in between. Each model consisted of an embedding linear layer, 6 steerable convolutions followed by spherical quotient ELU and batch normalization, and an MLP. The initial layer took the normals of each point in the point cloud with the standard representation as input. We utilized steerable vector spaces up to order 2 in each convolutional layer. To ensure comparability, we only varied the number of channels in each layer, aiming for a similar overall number of parameters among the models. The number of channels for each layer and the total number of parameters for each model are provided in Table A3. The last layer generated a 128-dimensional vector comprising scalar features for each node. Global max pooling was applied to obtain a 128-dimensional embedding of the point cloud. We further

---

product of two Wigner-D matrices is well known to decompose as $D_l \otimes D_j \cong \bigoplus_{i=|l-j|}^{l+j} D_i$. By applying this rule recursively, one can show that $D_1^{\otimes l}$ contains precisely one copy of $D_l$. We define $Y_l(x)$ as the linear projection of $x^{\otimes l}$ to this subspace. Note also that, since this is a linear projection, the definition of $Y_l$ satisfies the defining property of *homogeneous polynomials* $Y_l(\lambda x) = \lambda^l Y_l(x)$.

employed a 2-layer MLP ($128 \xrightarrow{ELU} 128 \xrightarrow{ELU} 40$) to calculate the class probability. In each convolutional layer, the point cloud was downsampled, resulting in the following sequence of input points: $1024 \rightarrow 256 \rightarrow 64 \rightarrow 64 \rightarrow 64 \rightarrow 16 \rightarrow 16$.

For the experiment results presented in Table 1, we scaled up the $SO(2)$-equivariant model by increasing the number of layers in implicit kernels to 3. We also used 6 convolutional layers with 20 channels each and employed the following downsampling: $1024 \rightarrow 256 \rightarrow 128 \rightarrow 128 \rightarrow 128 \rightarrow 64 \rightarrow 64 \rightarrow 64$. Additionally, skip connections were added between the layers, maintaining the same number of input points.

We trained each model using batch size 32 for 200 epochs, which we found to be sufficient for convergence. We minimized the cross entropy loss with label smoothing [48]. The position of

Table A3: Number of channels in each convolutional layer of a steerable CNN (see section 5.3).

| G | kernel | channels | #par, $\cdot 10^3$ |
|---|---|---|---|
| $M$ | Implicit | 20 20 20 20 20 128 | 122 |
| | Standard | 20 20 20 20 20 128 | 144 |
| $Inv$ | Implicit | 20 20 20 20 20 128 | 122 |
| | Standard | 20 20 20 20 20 128 | 144 |
| $SO(2) \rtimes Inv$ | Implicit | 11 11 12 12 12 128 | 588 |
| | Standard | 20 25 25 30 30 128 | 557 |
| $SO(2)$ | Implicit | 15 15 20 30 30 128 | 565 |
| | Standard | 30 30 40 40 40 128 | 561 |
| $SO(2) \rtimes F$ | Implicit | 12 12 12 12 12 128 | 580 |
| | Standard | 20 25 25 30 30 128 | 557 |
| $SO(2) \rtimes M$ | Implicit | 15 20 20 20 20 128 | 592 |
| | Standard | 30 40 40 40 40 128 | 592 |
| $SO(3)$ | Implicit | 10 10 10 10 20 128 | 160 |
| | Standard | 30 30 30 30 30 128 | 154 |
| $O(3)$ | Implicit | 10 10 10 10 20 128 | 140 |
| | Standard | 30 30 30 30 30 128 | 128 |

each point in a point cloud is normalized to the interval $[-1, 1]^3$ during the preprocessing. We used AdamW optimizer [30] with an initial learning rate $10^{-3}$. We also decayed the learning rate by $0.5$ after every 25 epochs. The training time on average was 3h on an NVIDIA Tesla V100 GPU and varied across different $G$. For the experiment reported in Table 1, the training time was around 30 hours on the same single GPU.

**QM9** During the preprocessing, we normalized the target variable by subtracting the mean and dividing it by the standard deviation computed on the training dataset. Each model has the following structure: embedding layer $\rightarrow$ steerable convolutional layers $\rightarrow$ global mean pooling $\rightarrow$ 2-layer MLP. The embedding layer consists of 3 parts: linear layer $\rightarrow$ learnable tensor product $\rightarrow$ spherical quotient ELU. First, it takes one-hot encoding of an atom type and applies a linear transformation. The learnable tensor product computes the tensor product of each field with itself to generate an intermediate feature map. Then, a learnable linear projection is applied to the feature map, which yields a map from trivial representations to spherical representations up to order $L = 2$. In the next step, a sequence of convolutional layers with skip-connection and quotient ELU non-linearities is applied. We use steerable feature vectors with a maximum order of 2 in each convolutional layer. The final classification MLP is defined as follows: $128 \xrightarrow{ELU} 128 \xrightarrow{ELU} 1$.

For the flexibility experiment, we employ steerable CNNs with 11 convolutional layers with residual connections and 24 channels. For the final performance indicated in Table 2, we scale the model up and increase its overall width and length. Implicit kernels are parameterized with $O(3)$-equivariant MLP with 3 linear layers with 16 fields and spherical quotient ELU in between. Concerning the number of parameters, the model with standard steerable kernels had 140k parameters, 1-layer $G$-MLP - 356k parameters, 2-layer $G$-MLP - 1.1M parameters, 3-layer $G$-MLP - 1.2M parameters.

We optimized the number of layers and channels for the $\varepsilon_{HOMO}$ regression task and the number of training epochs for the $G$ regression task. We trained each model using batch size 128 for either 125 epochs (the flexibility experiment) or 250 epochs (the final experiment). Each model is optimized with AdamW optimizer with an initial learning rate of $5 \cdot 10^{-4}$. We use learning rate decay by $0.5$ every 25 epochs. It takes around 20 minutes per epoch on an NVIDIA Tesla V100 GPU.

## B.3 Depth-width trade-off.

Romero *et al.*[39] indicated that implicit representation of convolutional kernels allows one to build a shallower model compared to standard CNNs. We, however, did not obtain a similar pattern.

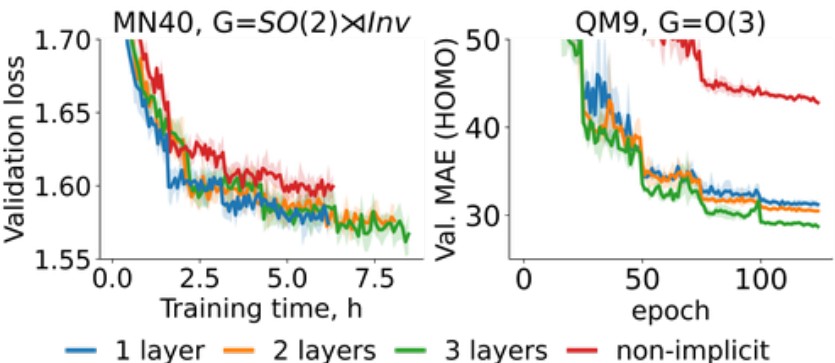

Figure 5: Learning curve for different steerable kernels.

Even though the width of standard steerable convolutional kernels must be pre-specified, keeping it sufficiently large yields the same scaling pattern as for implicit ones. We note that implicit kernels can adapt their width and thus the field of view, which is not the case for standard steerable kernels. We hypothesize that the result might change on different datasets where long-range dependencies play a more important role, e.g. in sequential data, as shown in [39].

### B.4 Complexity and training time

Overall, implicit kernels offer a flexible alternative to standard steerable kernels, potentially at an increased computational cost compared to optimized implementations of handcrafted steerable bases. However, the additional cost depends only on the MLP's complexity and can be controlled during the design. We demonstrate the effect of the MLP's depth on performance and training time in Fig. 5. In practice, we find that two layers provide the best trade-off, but increased complexity might be beneficial when including additional attributes (Fig. 5, right). We also demonstrate that training and inference time are approximately equal for standard steerable kernels [53] and implicit kernels parameterized by a single layer. Finally, we only experienced training challenges (e.g. instabilities) when using MLPs with #layers > 3. To overcome these issues, we used batch normalization of spherical harmonics, i.e. the input of MLPs, which proved to be effective.

Importantly, implicit kernels don't have substantially more hyperparameters compared to the method suggested in [7]. One only has to tune the parameters of a $G$-MLP, which are arguably easier to interpret than the parameters of handcrafted bases that are typically group-specific.