# OpenReview forum: "Implicit Convolutional Kernels for Steerable CNNs"
_NeurIPS.cc/2023/Conference — NeurIPS 2023 poster_

### Official Review · Reviewer_TwdU · 2023-06-29

**Soundness:** 3 good
**Presentation:** 3 good
**Contribution:** 3 good
**Rating:** 6
**Confidence:** 3

**Summary:**

This paper tackles one problem of Steerable CNN: one needs to analytically solve a group G-specific equivariant constraint (eq 2 in the paper) in order to obtain the basics for the kernel.

The authors propose to avoid this analytical solution by using a G-equivariant MLP to parameterize a G-steerable kernel basis (hence called implicit kernel). By doing so, they obtain a framework that allows one to achieve equivariance to various groups even when an analytical solution for the kernel basis is not know.

The authors provide a theoretical proof that MLP equivariance is sufficient for building an equivariant steerable convolutional layer and empirically show that the suggested framework is useful in practice. Specifically, they show the relevance of the model being able to adapt to smaller G < O(n)  (as opposed to assuming O(3) as in SEGNN) using the N-Body simulation system. They show the generalizability of their method on the ModelNet-40 dataset where the proposed technique works better than Standard Steerable CNNs when the analytical solution for the basis is not available. And finally they show the flexibility of their method that allows one to introduce additional constraints (such as bonds between molecules or atoms) on QM9 dataset. In both the last two tasks the authors compare their method with an extensive list of SOTA methods.


**Strengths:**

- The paper tackles an important limitation of steerable CCNs.
- The proposed solution is shown to be effective, at least when the analytical solution is unknown. Hence constitutes a detectable and positive improvement for the community.
- The paper reads well.


**Weaknesses:**

In my opinion there is an opportunity to strengthen the paper with a bit more empirical analysis. Since the kernel is learnt, it might be insightful to know to which extent the learnt solution is equivariant. Currently accuracy on downstream tasks is used as a proxy for that, but a specific analysis on equivariance violation could further enhance the paper. For example, is there a relation with the MLP size and the ability to learn the equivariance? Etc...

Some parts could be more clear or more details could be provided. There are missing details about some experimental tasks, or how the model size used compares to the SOTA, etc… Please see the questions below for a more concrete list.

The paper is not accessible to the wide ML community. Specifically the paper makes a lot of assumptions about the reader’s knowledge. I don’t think this is a reason for rejection but it would be great to make an effort, whenever possible, to make the paper more accessible. Just a few examples but there are many through the paper:
- not all readers might be familiar with the concept of natural action (line 82). I’d recommend to clarify it.
- "matrix containing the Clebsch-Gordan coefficients". Could explain what they are or add a reference.
- Some of the notation used could be clarified, for example not everyone is familiar with the outer semidirect product used. It would be great to define it.

**Minor notes:**
- Definition 3: I’ would consider adding “for all g in G in x in X”
- Line 289 “it’s” ==> it is (to be consistent with the rest of the paper)
- Line 314 refers to Table 5.3 but I think it might be Table 1?
- In the Bibliography the name of conferences are not consistent: sometimes it uses the full name of the conference  and the abbreviation, some other just the abbreviation, other again just the full name.


**Questions:**

I think the paper is good. There are opportunities to strengthen it by addressing some of the weaknesses mentioned above and by addressing the questions here.

- Do the authors think it’s useful and possible to provide the empirical analysis mentioned in the weaknesses.
- Can the author make an effort to make the paper more accessible?
- In section 2 the authors write “Trivial representation: all elements of G act on V as the identity mapping of V , i.e. ⇢(g) = I. “
This does not seem to satisfy the definition 2, specifically this mapping does not seem to be invertible. Is this statement correct?
- Can you describe maybe with an example how you would build the G-equivariant MLP? Section 3.4 would benefit from a more concrete explanation (even if in the appendix). Even better if the authors would consider to publicly releasing the code.
- “We choose the model’s hyperparameters to have a similar parameter budget to the SEGNN”. Can one be more specific? How many parameters for example were trained in SEGNN compared to the proposed model?
- In the ModelNet-40 experiment there is a sampling step involved. The Figure reports results with error bars using 5 runs but it’s unclear if the 5 runs are with different initial seeds or different sampling. Also is 5 runs sufficient?
- In Figure 2: Do the authors have any intuition about why there is such a large STD for G=M but not for other groups?
- 5.3: The task for this experiment could be explained better. The dataset is clear but what is the task?
- QM9 task: “Both models we reported here have approx. 2 · 10^6”  How does this compare to other models?
- For the QM9 tasks more details about the meaning of the regressed energy values would help. Not everyone if familiar with this dataset and task.

**Appendix B2**
- For the N-Body “The run-time, on average, was 5 min. “ On which machine? And what is “run-time”? Per Single point? Per Bach? For the whole training?
- For  ModelNet-40 the machine is specified but the what is ”Runtime” should still be clarified.

**Limitations:**

The papers states the limitations in within the results discussion. For example :
- ”The only statistically significant negative difference is presented for G = SO(3), for which a tailored and hence optimal kernel basis is already available.”
or
- “Although we do not outperform more task-specific and involved approaches such as PointMLP [31],”
or
- "For the remaining energy variables (G, H, U, U0, ZPVE) and R2, our model significantly falls behind the task-specific benchmark approaches such as DimeNet or PaiNN.”

Additionally there is a limitation section in Appendix C. I would encourage the authors to bring the limitation section from the appendix into the paper and summarize the limitations found during all the experiments in the same section.

---

> ### Author Rebuttal · Authors · 2023-08-08
>
> We appreciate the reviewer found our contribution to be a positive improvement for the community. We will address the reviewer's questions and suggestions one by one.
>
> ### Weaknesses
> - We would like to emphasize that the learned kernels are already equivariant to a pre-defined group $G$ by construction (see Lemma 1). Some design choices, e.g., non-linearity, might loosen the exact equivariance, but the problem is characteristic of the framework of Steerable CNNs as a whole. We also emphasize that we do not learn the group $G$ - it is instead a hyperparameter of the model.
> - We will include more details concerning the computational complexity and memory cost of all models in the supplementary material of the camera-ready version.
> - We agree with the reviewer that some concepts can be further clarified. We will use the additional space of the camera-ready version to include additional theoretical details and make the manuscript more accessible.
>
> ### Questions
> - We address the question in point 1 of weaknesses.
> - Definitely. We will add theoretical details on steerable CNNs in the camera-ready version to improve accessibility as well as improve the flow to improve readability.
> - We specify that a representation $\rho(g)$ of a group $G$ should map into the set of invertible matrices rather than be faithful as the reviewer suggests. Therefore, the constraint is satisfied for trivial representations, as $\rho(g) = I$.
> - We will indeed release the code with the camera-ready version. We also aim to expand section 3.4 to make it more accessible. It is worth mentioning that the procedure described in section 3.4 is already implemented in public libraries, for example, $\texttt{escnn}$.
> - Canonical SEGNN model has $147$K parameters and the proposed model has $93$K parameters.
> - We use a different seed for each run. Five runs is a typical choice in deep learning experiments to estimate statistics. Moreover, given the low standard deviation observed, we find that five runs were sufficient in practice.
> - While it is hard to give a precise answer, we hypothesize that this happens since $G = M$ and $G = Inv$ lead to much less constrained models compared to other groups since those groups are very small (each has $2$ elements). Furthermore, learned kernels might vary significantly, leading to high standard deviation.
> - The task is to classify point cloud models. ModelNet-40 has 40 classes of furniture. In our experiments, we evaluate the average accuracy across all classes.
> - For comparison, SEGNN has $1 \cdot 10^6$ parameters, PaiNN has $0.6 \cdot 10^6$ parameters, SphereNet, DimeNet, and GemNet all have around $2 \cdot 10^6$ parameters.
> - We agree that more detailed information regarding the QM9 dataset should be included and will add a brief description of the variables in the supplementary material.
> - The training was performed on 1 NVIDIA Tesla V100 GPU, and the run-time refers to the whole training. We will include those clarifications in the supplementary material.
> - The run-time (training time) varied across different $G$. On average, it took $4$ to $8$  hours for a model to converge.

---

> > ### Comment · Reviewer_TwdU · 2023-08-10
> > **Thanks for considering my suggestions and addressing my questions**
> >
> > I would like to thanks the authors for answering my questions. I am satisfied with the answers and I believe that if the authors will include the modifications suggested in the manuscript the paper will be stronger.
> >
> > The only two comments (none strictly needed for a weak acceptance but something for the authors to consider) that I would like to add are:
> >
> > 1. On the first weakness: as the authors mentioned, there is a gap between the theory and the practical implementation. What I think would add value is to test to which extent those design choices ( e.g., non-linearity) have loosen the exact equivariance.
> > 2. On the statement "Five runs is a typical choice in deep learning experiments". I believe this is an arguably bad choice, especially in a case (like this one) where there are two sources of stochasticity (random seed and sampling step). I believe that a more statistically significant analysis would have made the empirical evaluation stronger (but again not needed for weak acceptance).

---

> > > ### Author Response · Authors · 2023-08-16
> > >
> > > 1) In this work, we use quotient nonlinearities (see [7] for details) that employ discretised Fourier transform, which renders them approximately equivariant. We think the effect of such nonlinearities is beyond the scope of this paper but has been actively studied in a few previous works. See, for example, [a], [b], [c]. The key finding is that by keeping the number of samples sufficiently high, one can achieve a negligibly small equivariance error. In our experiments, the equivariance error (MSE) was approximately $10^{-5}$.
> > > 2) We would like to emphasise that the variance at five samples is particularly low for most groups in the MN-40 experiment, so we don’t expect the average performance to change much with more samples.
> > >
> > > [a] Franzen, D., & Wand, M. (2021). General Nonlinearities in SO(2)-Equivariant CNNs. Neural Information Processing Systems.
> > >
> > > [b] Poulenard, A., & Guibas, L.J. (2021). A functional approach to rotation equivariant non-linearities for Tensor Field Networks. 2021 IEEE/CVF Conference on Computer Vision and Pattern Recognition (CVPR), 13169-13178.
> > >
> > > [c] Cohen, T., Geiger, M., Köhler, J., & Welling, M. (2018). Spherical CNNs. ArXiv, abs/1801.10130.

---

> > > > ### Comment · Reviewer_TwdU · 2023-08-21
> > > >
> > > > Thank you for your response.

---

### Official Review · Reviewer_GuaE · 2023-07-06

**Soundness:** 4 excellent
**Presentation:** 4 excellent
**Contribution:** 3 good
**Rating:** 7
**Confidence:** 3

**Summary:**

This work considers the setting of steerable networks, in which a particular kind of group-equivariant (alternatively, G-steerable) kernel is translationally convolved with an input vector (or matrix) field. The group-specific design of such a network focuses on the derivation of the group-equivariant kernel. Although prior work laid out a procedure for deriving an analytic, orthogonal basis of functions whose span encompasses all such G-steerable kernels, this paper presents a conceptually simpler method: parametrizing the kernel with an equivariant map. This method is flexible and allows conditioning the kernel on other invariant or equivariant attributes. They test their method on datasets of point cloud classification, synthetic N-body simulations, and molecules (QM9).

**Strengths:**

Originality: Using an equivariant network to parametrize a group-steerable kernel is a natural and elegant idea. Although implicit kernels have been used in other settings, their application to steerable kernels is novel and well-founded. Conceptually, it is simpler than previous analytical approaches to deriving explicit, orthogonal bases.

Clarity: The proposal is clearly described, and the paper provides a helpful consolidation of related work overall.

Quality: The proposal is indeed equivariant, and experiments show good results over a sufficiently wide range of settings.

Significance: Group-steerable networks are widely used, and this formulation seems to apply to all compact groups of interest.

**Weaknesses:**

1. It seems to me that the vast majority of application problems involving subgroups of $E(3)$ acting on $\mathbb{R}^3$ simply involve $SO(3)$ or $O(3)$. In that sense, the scope of this paper may be limited. Nonetheless, this does not preclude other groups of interest from arising in future applications, but the paper could be strengthened by providing additional motivating examples.
2. The tradeoffs between this method and the analytical method of [7] are subtle, and indeed they require many of the same tools (for example, knowing the irrep structure of the group of interest, so that one can construct equivariant functions over the input space, even if they are not required to be orthogonal). As noted in the questions, I do not fully understand the benefits of this method as compared to [7], although it is clearly more general and simpler to articulate.

**Questions:**

1. My biggest question is that I do not understand the second paragraph of Section 3.2, comparing this approach with that of [7]. The paragraph seems to claim that the basis produced by [7] might be too large, but I thought [7] would produce an orthogonal equivariant basis — and if it is orthogonal, how can the “size” of the basis be suboptimal (as it is determined the dimension of the vector space of steerable functions)?
2. On a related note, both this paper and [7] seem to require knowledge of the irreps of the group of interest. Can the authors elaborate on which of the ingredients (a) through (d) listed at the bottom of page 5 of [7], are troublesome? Section 3.2 seems to imply not that the subgroup method for obtaining a steerable basis is computationally intensive, but that it produces a lower quality output — is that the case?
3. If already provided with a steerable basis in the style of [7], could one still “condition” on physical info (at least invariants) by just storing a bank of coefficients, and choosing different coefficients based on endpoint features? If so, this could make more sense as a baseline experiment. Of course, such a method may not choose coefficients smoothly as a function of the endpoint features, which is one potential benefit of the authors’ proposed conditioning method.
4. To clarify, if you had a G-equivariant MLP, is the primary barrier to using this G-MLP to derive a G-steerable orthogonal basis the orthogonalization? In other words, one could set random weights to the G-MLP, and then try to orthogonalized to get a basis of equivariant functions? (I think the problem is this: orthogonalization can be done via Gram-Schmidt, but only if one can compute inner products exactly — with $\mathbb{R}^3$ as the base space, it does not seem like one could do this exactly. However, I would be happy to hear the authors’ thoughts on this, as if there were an easy way to reduce a G-equivariant MLP directly to a steerable basis, it would perhaps weaken the argument that their proposed method is easier than computing an explicit G-steerable basis.)
5. In Section 5.2, why is the baseline an $O(3)$-equivariant net? This is an unfair comparison for a problem that is not $O(3)$-equivariant, since such a network can never express the ground truth solution. Can the authors report performance relative to a generic net instead (e.g. only translation invariant), or an approximately equivariant network?
6. What do the authors hypothesize causes the difference in performance shown in Figure 3? For example, is it conditioning on radial vectors?

As a quick note, here is one other piece of slightly related work at the intersection of equivariance and implicit convolutional kernels: “Relaxing equivariance constraints with non-stationary continuous filters” by Ouderaa et al 2022.

**Limitations:**

There are not potential negative societal impacts.

---

> ### Author Rebuttal · Authors · 2023-08-08
>
> We are happy the reviewer found our idea elegant and well-founded and appreciated the clarity of the presentation. We now would like to address the concerns that were raised.
>
> ### Weaknesses
> - We would like to emphasize that 2 out of 3 experiments deal with symmetry groups other than $E(3), O(3), SO(3)$, namely MN40 and N-body problems, where $G = SO(2)$ or $O(2)$. However, we agree that in future, there might be applications that would find our approach particularly relevant. For example, we find that the task of molecular binding is particularly relevant, which we try to imitate with the N-body experiment. Besides, the generalization to pixel and voxel data is quite straightforward, for which our work provides the necessary theoretical foundation.
> - We appreciate the reviewer's question regarding the comparison between our method and [7]. We would like to note that the benefits of our approach are 1) better performance for the same $G$ (see MN40 experiments), 2) enabling conditioning (see QM9 experiments), and 3) overall simplicity of implementation. To better highlight the difference, we now provide a table (see the PDF file attached to the global rebuttal) with key ingredients for both approaches. The key difference is that [7] requires an ad-hoc $G$-steerable basis, which might not be trivial to develop while we alleviate it. Additionally, in [7], authors use group restriction, which has its own limitations described in 3.2.
>
> ### Questions
> - To clarify the difference with [7], we will further expand the section in the camera-ready version. Besides, we now include the requirement table, which might be of help. Regarding the question: the $G$-steerable basis used by [7] is only a finite subset of the analytical basis for the infinite-dimensional space $L^2(\mathbb{R}^n)$. However, because the space spanned by the basis must be closed under $G$, the choice of $G$ limits the degree of discretization of the analytical basis. Hence, group restriction might lead to a suboptimal basis because the $G$-optimal discretization might not be optimal for another group $G' \subset G$.
> - In our opinion, the most troublesome ingredient from [7], section 3, is *b*, the $G$-steerable basis $\mathcal{B} = \{Y_{ji}\}_{ji}$ for $L^2(\mathbb{R}^n)$), which we tackle in the paper.
> - We find that alternative viable yet limited. The main concern is that the strategy is only feasible for discrete invariant variables, while continuous invariant variables require ad-hoc adaptations. One also has to ensure equivariance when dealing with $G$-equivariant features, which might be far from trivial as it is within our framework.
> - While the approach suggested by the reviewer might be a valid initialization scheme for an implicit steerable kernel, we want to emphasize that the issue with a $G$-steerable basis doesn't root in the orthogonality but rather in its finiteness (see p.1 in Questions).
> - We agree with the reviewer that a non-equivariant baseline would be appropriate and, furthermore, include it in the experiment. Below we indicate the performance (MSE averaged over 5 runs) of a message-passing neural network (a non-equivariant counterpart of SEGNN) with SEGNN and our model for reference:
>
> |   Stiffness  |      0     |      1     |      5     |     10     |     25     |     50     |     100    |     200    |    1000    |
> |:------------:|:----------:|:----------:|:----------:|:----------:|:----------:|:----------:|:----------:|:----------:|:----------:|
> |     MPNN     |   0.0022   |   0.0031   |   0.0068   |   0.0087   |   0.0030   |   0.0162   |   0.0560   |   0.0978   |   0.1065   |
> |  O(3)-SEGNN  | **0.0009** |   0.0092   |   0.0117   |   0.0183   |   0.0291   |   0.0151   |   0.0229   |   0.0938   |   0.1313   |
> | SO(2)-CNN-IK |   0.0010   | **0.0010** | **0.0009** | **0.0008** | **0.0008** | **0.0008** | **0.0014** | **0.0043** | **0.0162** |
>
> - We indicate that Fig.3 depicts the benefit of using more flexible implicit $G$-steerable convolutions instead of ones using the approach described in [7]. We argue that the difference is caused by the optimality of implicit basis (see p. 1 in Questions) (sub-optimal for [7], task and $G$-specific for ours).
> - We are happy to include the paper mentioned since we find it indeed highly related.

---

> > ### Comment · Reviewer_GuaE · 2023-08-20
> > **Thank you for the response**
> >
> > Thank you to the authors for their thoughtful response. I am inclined to raise my score, but I have just a few remaining questions and comments.
> >
> > 1. Thank you for the new experiment with a non-equivariant baseline. I am satisfied with these results.
> > 2. Unless I am mistaken, Lemma 1 is a basic linear algebraic rearrangement of Equation 2. Solving one seems equivalent to solving the other. Therefore, the response to reviewer fqKN’s first question does not make sense to me.
> > 3. Do the advantages of this work hinge on the assumption that the equivariant MLP is a universal approximator?
> > 4. In the case of SO(3), the spherical harmonics weighted by radial basis functions provide a G-steerable basis for SO(3) [3D steerable CNNs, Weiler et al 2018]. To what extent (if any) does this phenomenon generalize to other groups? I find this important to check, because I would like to clarify whether knowing an analytic form for the irreducible representations of a group can provide a G-steerable basis.

---

> > > ### Author Response · Authors · 2023-08-21
> > >
> > > 2. It is correct - the two constraints are equivalent. However, the solution strategy in [7] requires an explicit steerable filter basis (see the requirement table in the global rebuttal). Concerning our approach, the algebraic rearrangement we employ is convenient because it allows us to replace the manually constructed (ad-hoc) basis with a generic $G$-MLP.
> > > 3. Yes, it is correct.
> > > 4. Spherical harmonics are not the irreps of SO(3); they are steerable functions over the 2-sphere. The analytical solution for $SO(3)$ [3D steerable CNNs, Weiler et al. 2018] implicitly assumes 1) decomposition of $R^3$ into orbits of $SO(3)$, i.e. as a product of $S^2 \times R+$, 2) a steerable basis for each orbit (i.e. the spherical harmonics) 3) a basis across the orbits (i.e. the radial component). In general, knowing the irreps of a group is not sufficient to build a steerable basis.

---

> > > > ### Comment · Reviewer_GuaE · 2023-08-21
> > > > **Thanks for clarifying**
> > > >
> > > > I will upgrade my score to "accept" under the assumption that the authors will incorporate the changes discussed by reviewers; particularly, the more detailed explanation of the advantages of this method over that of [7]. I think this work, although addressing a perhaps niche problem, could be useful for the broader community and is a natural use of implicit convolutional kernels.

---

### Official Review · Reviewer_fqKN · 2023-07-08

**Soundness:** 3 good
**Presentation:** 3 good
**Contribution:** 3 good
**Rating:** 6
**Confidence:** 3

**Summary:**

The paper proposed implicit neural representations via MLPs to parameterize G-steerable kernels. Additionally, the performance of the approach is extensively empirically tested against alternative methods on multiple tasks.


**Strengths:**

- Conceptually

In many cases, steerable group equivariant neural networks yield more accurate equivariance in practice compared to counterparts with regular representations that use sampling. Implicit kernels have shown to be beneficial in the space of regular group equivariant networks but have not yet been explored in the context of steerable equivariance. This paper demonstrates how steerable equivariant networks can also be extended to use implicit kernels.

- Scientifically

The paper extensively validates the method on actual datasets in which more complex symmetries play an important role. The paper extensively validates the approach on both graph structures and dynamical systems, demonstrating relevance in real-world applications and also includes extensive comparison with existing baselines.

**Weaknesses:**

- Implicit kernels do not alleviate the need to solve Eq.2 analytically.

The method relies on steerable kernels, which require analytically solving Eq. 2 for specific group G of interest. Implicit kernels do not alleviate this, as the constraint of Eq. 2 must still be solved in order to build G-equivariant MLPs. Although these are shortcomings of steerable equivariance and not of the implicit kernels, it should be stressed that implicit kernels do not mitigate these shortcomings.

- High computational complexity and memory cost

The method only seems to increase computational and memory complexity over already more expensive steerable kernels.

- No benefit with increased depth of implicit kernels

The paper relies on using multi-layer MLPs as implicit kernels but notices that deeper MLPs do not necessarily lead to improved performance (App. B.3 and App. B.4). The paper does argue that the increased flexibility that implicit kernels offer may benefit long-range dependencies but does not conduct any experiments on this. It would be interesting to demonstrate the benefit of implicit kernels on tasks where deeper kernels help.

- Class of implicit functions

Is there a reason why only multi-layer perceptrons are considered to parameterise implicit functions? Since the depth of implicit functions has not been found to be beneficial, it could also be interesting to consider different simpler non-linear bases. Even more so, because they appear to be effective in the context of regular group equivariant networks (e.g. sinusoidal/fourier [1], b-splines [2] or simple radial basis features [3]). Or is there a reason why this would not be applicable or less interesting in this setting?

[1] Sitzmann, Vincent, et al. "Implicit neural representations with periodic activation functions." Advances in neural information processing systems 33 (2020): 7462-7473.

[2] Bekkers, Erik J. "B-spline cnns on lie groups." arXiv preprint arXiv:1909.12057 (2019).

[3] van der Ouderaa, Tycho FA, and Mark van der Wilk. "Sparse Convolutions on Lie Groups." NeurIPS Workshop on Symmetry and Geometry in Neural Representations. PMLR, 2023.


**Questions:**

a) Am I correct in the assessment that the method does not alleviate the need to solve Eq. 2 analytically and thus does not directly improve the efficiency of steerable kernels in that regard?

b) What are computational runtimes? (and what is the explicit overhead of implicit kernels)

c) What are memory costs? (and what is the explicit overhead of implicit kernels)



**Limitations:**

The paper extensively compares (externally) with alternative approaches. However, the (intrinsic) analysis of how different types of implicit kernels may affect performance remains limited. One of the main benefits of implicit kernels is the increased flexibility it provides when designing the model. It would therefore be very interesting to get a better idea of how different choices in the functional form of the implicit kernels affect the output or model performance. This point becomes especially relevant since the paper includes negative results on deeper implicit kernels (which nevertheless are interesting findings), as this would help demonstrate how implicit kernels can be beneficial in practice.

---

> ### Author Rebuttal · Authors · 2023-08-08
>
> We thank the reviewer for highlighting the scientific and conceptual strengths of the manuscript. We have carefully considered your comments and suggestions and provide a detailed rebuttal below.
>
> ### Weaknesses
> - We want to emphasize that implicit kernels do allow us to alleviate solving Eq.2 by relaxing the constraint to $G$-equivariance of the implicit representation (see Lemma 1). In particular, one does not need anymore to manually construct *ad-hoc* $G$-steerable basis, which otherwise comes from solving Eq.2. To summarize the difference between implicit kernels and standard steerable kernels, we now provide a table (see the PDF attached to the global rebuttal) with key ingredients for both approaches.
> - We would like to highlight that the complexity of our approach depends entirely on the structure of $G$-MLPs which is defined by the user. In preliminary experiments on MN40, we observed that 1-layer $G$-MLP achieves results better than the non-implicit kernels suggested in [7] while having a comparable number of parameters (approx $2\cdot10^4$). We provide a more detailed analysis in Appendix B.4.
> - We analyze the effect of depth in the Appendix (see Fig. 5). For the QM9 dataset, and to a lesser extent for MN40, we observe quantitative improvement by increasing the number of layers in $G$-MLP.
> - We agree with the reviewer that employing our approach on data exhibiting symmetry and long-range dependencies would be an interesting direction, which we leave for further work. Currently, preliminary experiments do not allow us to draw conclusions as long-range dependencies are practically absent in the data we work with.
> - We would like to emphasize that the depth of $G$-MLP does matter according to the evidence from our experiments. Concerning the choice of $G$-MLP, there is indeed no restriction on using different implicit representations of kernels. However, applying Fourier/B-splines/etc. to achieve $G$-equivariance is similar to crafting a custom $G$-steerable filter basis. In contrast, we intended to develop a more general framework employing generic implicit $G$-MLPs to model general convolution kernels. Nevertheless, we acknowledge the reviewer's suggestion of investigating alternative implicit functions to enhance the computational efficiency of implicit kernels, which could be a promising avenue for future research.
>
> ### Questions
> - No, see point 1 in weaknesses.
> - We report the training time implicitly in Fig.5, where models are trained for the same number of epochs. While deeper $G$-MLPs are indeed more computationally expensive than non-implicit kernels, 1-layer MLPs achieve comparable run-time.
> - Let us estimate the computational cost as the amount of memory required to store the model's parameters, activations, and intermediate tensors during forward and backward propagation. According to our evaluation, all models with $G=SO(2)$ applied to MN40 had comparable amounts of parameters ($120$k) and consumed approximately the same amount of memory - $1.5$ MB regardless of the architecture when processing a batch of size $16$. Considering models in the QM9 experiment, the ones with standard steerable kernels had $140$k parameters, 1-layer $G$-MLP - $356$k parameters, 2-layer $G$-MLP - $1.1$M parameters, 3-layer $G$-MLP - $1.2$M parameters. We will include those details in the supplementary material of the camera-ready version.

---

### Official Review · Reviewer_AqiD · 2023-07-09

**Soundness:** 4 excellent
**Presentation:** 3 good
**Contribution:** 3 good
**Rating:** 6
**Confidence:** 3

**Summary:**

This work proposes a novel method for achieving a G-equivariant neural network by implicitly parameterizing the steerable filters by G-equivariant MLP instead of learning them with a steerable basis function. The proposed frame is flexible, and the implicit kernel can also consider the problem context by expanding input to the kernel with problem-related parameters. The work provides theoretical justification for the proposed method and shows the model's effectiveness via empirical study.

**Strengths:**

1. The work proposes a novel method to achieve G-equivariant by implicit parameterization, which provides more flexibility to the G-equivariant model. And empirically shows the benefit of the flexible kernel that can consider the additional problem-dependent features.
2. The text is easy to follow and clearly written

**Weaknesses:**

1. Evaluation: The empirical evaluations are performed on N-body simulations, ModelNet, and the QM9 dataset. Which is different from the previous works [1,2,3]. And this makes it difficult for readers to make a direct comparison between the approaches. And raises the question of whether the performance of the approaches is significantly dependent on the chosen problem.
2. The proposed method utilized G-equivariant MLP for implicit parameterization of the kernel. But the rationale for choosing an implicit steerable kernel over EMLP[4] is not addressed nor included in the evaluations.

1.Steerable CNNs

2.3D Steerable CNNs: Learning Rotationally Equivariant Features in Volumetric Data

3.Clebsch–Gordan Nets: a Fully Fourier Space Spherical Convolutional Neural Network

4.A Practical Method for Constructing Equivariant Multilayer Perceptrons for Arbitrary Matrix Groups

**Questions:**

1. The G-equivariant MLPs also require us to solve constraints [1]. And we need to do the same for the steerable methods. How are these two different?
2. The result of Table 1: What is the rationale for choosing a problem where we have non-G-equivariant models outperforming G-equivariant methods?
3. Line: 105, equation 2 seems a little out of context. Even though the relevant work is cited and a little background on how we reach that constraint will make the work self-contained.

1.A Practical Method for Constructing Equivariant Multilayer Perceptrons for
Arbitrary Matrix Group

---

> ### Author Rebuttal · Authors · 2023-08-08
>
> We are delighted that the reviewers acknowledge the novelty and flexibility of our proposed method as well as the clarity of our manuscript. Below, we address the reviewers' concerns point-by-point:
>
> ### Weaknesses
> - We would like to emphasize that each of the approaches mentioned by the reviewer deals with distinct data modalities ([1]: pixel data, [2]: voxel data, and [3]: point cloud data). As our paper primarily focuses on the point cloud data, it is most appropriate to compare it to approach [3], which also utilizes similar datasets: ShapeNet (we employ MN-40) and QM7 (we use QM9). Although our dataset choices differ from this specific case, they are widely adopted and considered standard in current literature (see, for example, [34, 5]).
> - While the paper did not explicitly address the topic, we will incorporate a discussion in the related work section. The primary rationale behind this decision is that EMLP supports equivariant MLPs but not convolutions, rendering it incapable of directly processing image/volumetric data in the way Steerable CNNs do. Henceforth, we focus on CNNs with consideration for potential extensions to various data modalities.
>
> ### Questions
> - We acknowledge the reviewer's concerns about different types of constraints arising when dealing with Steerable CNNs. To differentiate, we present a summary table (see the PDF file attached to the global rebuttal) outlining the essential components for constructing steerable kernels in our approach compared to the baseline method from [7]. In short, solving the constraint from [1] involves obtaining a $G$-steerable basis, whereas building $G$-MLP does not.
> - The rationale was to choose a dataset with a symmetry group $G \neq SO(3)$ (for which MN40 suffices since every object is vertically aligned), so we can explore how the choice of $G$ affects the performance of $G$-equivariant models rather than try to achieve the state-of-the-art performance.
> - We understand the concern regarding the context of equation 2 in line 105. We will revise the presentation in this section in the camera-ready version to provide a more coherent flow.

---

> > ### Comment · Reviewer_AqiD · 2023-08-17
> > **Response**
> >
> > I thank the authors for their response.

---

### Author Rebuttal · Authors · 2023-08-08

We appreciate the reviewers’ thoughtful feedback on our submission and have carefully addressed each of the questions raised in separate responses. We are glad to hear that reviewers found the proposed approach of using implicit neural representation to parameterize steerable kernels novel and flexible, with potential benefits in integrating problem-dependent features. Additionally, we appreciate their recognition of the extensive validation and comparisons with existing baselines and the clarity and significance of our work in the context of Steerable CNNs.

A recurrent concern was that the difference in hardness of implementation compared to the baseline [7] was unclear. To address the issue, we now provide a table (see the PDF attached) highlighting each method’s key ingredients and the comparative hardness of their implementation.

---

### Decision · Program_Chairs · 2023-09-21

**Decision:**

Accept (poster)

**Comment:**

Most reviewers find that using an equivariant network to parametrize a group-steerable kernel is an elegant idea, and all recommend acceptance after the rebuttal. AC recommends accepting the paper. However, AC encourages the authors to incorporate some outstanding comments raised by the reviewers. For example, as commented by Reviewer TwdU that the paper may not be easily accessible to the wide ML community, and some concepts should be further clarified, e.g., the concept of natural action. Also, in the authors’ response, “the generalization to pixel and voxel data is quite straightforward, for which our work provides the necessary theoretical foundation.”, AC finds it not obvious how the proposed method can be extended to, e.g., images. It will be outstanding if the authors consider additional explanations about the above statement.